# Strategies to Reduce the Use of Antibiotics in Fresh and Chilled Equine Semen

**DOI:** 10.3390/ani14020179

**Published:** 2024-01-05

**Authors:** Sonsoles Mercedes Zabala, Consuelo Serres, Natalia Montero, Francisco Crespo, Pedro Luis Lorenzo, Verónica Pérez-Aguilera, Carmen Galán, Mónica Domínguez-Gimbernat, Agustín Oliet, Santiago Moreno, Bruno González-Zorn, Luna Gutiérrez-Cepeda

**Affiliations:** 1Animal Medicine and Surgery Department, Veterinary Faculty, UCM, Avda. Puerta de Hierro s/n, 28040 Madrid, Spain; sonsoles.zabala@madrid.org (S.M.Z.); cserres@ucm.es (C.S.); fcrecas@oc.mde.es (F.C.); cgalan01@ucm.es (C.G.); monicadominguez@ucm.es (M.D.-G.); 2Animal Selection and Reproduction Center, Madrid Institute for Rural, Agricultural and Food Research and Development (IMIDRA), Ctra. Colmenar Viejo a Guadalix de la Sierra, km 1, Colmenar Viejo, 28770 Madrid, Spain; agustin.oliet@madrid.org (A.O.); santiago.moreno@madrid.org (S.M.); 3Animal Health Department, Veterinary Faculty, UCM, Avda. Puerta de Hierro s/n, 28040 Madrid, Spain; nmontero@vet.ucm.es (N.M.); bgzorn@ucm.es (B.G.-Z.); 4Centro Militar de Cría Caballar de Ávila (CCFAA), C/Arsenio Gutiérrez Palacios s/n, 05005 Ávila, Spain; veronicaperezaguilera@gmail.com; 5Physiology Department, Veterinary Faculty, UCM, Avda. Puerta de Hierro s/n, 28040 Madrid, Spain; plorenzo@ucm.es

**Keywords:** chilled equine semen, semen microbial load, simple centrifugation, single-layer colloidal centrifugation, filtration, absence of antibiotics, antibiotics-free extender

## Abstract

**Simple Summary:**

The escalating threat of antimicrobial resistance, identified by the World Health Organization as a top global public health concern, prompts a critical examination of practices contributing to this problem, as in the case of artificial insemination where antimicrobials are commonly included in sperm doses. This study assessed three equine semen processing techniques (Simple Centrifugation, Single-Layer Colloidal Centrifugation, and Filtration) for their impact on quality and microbial load, both immediately after processing and after refrigeration for 48 h. Results showed no significant differences in sperm quality immediately after processing among protocols, except for a higher straightness index in the filtrated and colloidal-centrifuged samples compared to raw semen. In 48 h chilled samples, only the linearity and oscillation index were significantly higher in colloidal-centrifuged samples. Microbial load analysis revealed no significant differences between protocols after refrigeration and minor differences between some protocols and raw sperm values. Thus, irrespective of antibiotic presence, the evaluated methods maintained semen quality and reduced microbial load to the same extent as the traditional antibiotic-containing protocol. These findings suggest the potential of alternative processing protocols, coupled with hygiene practices, to mitigate or eliminate non-therapeutic antibiotic use and contribute to controlling bacterial resistance. Further studies are warranted to validate these results.

**Abstract:**

The study assessed the impact of four equine semen processing techniques on sperm quality and microbial load immediately post-processing and after 48 h of refrigeration. The aim was to explore the potential reduction of prophylactic antibiotic usage in semen extenders. Semen from ten adult stallions was collected and processed under a strict hygiene protocol and divided into four aliquots: Simple Centrifugation with antibiotics (SC+), Simple Centrifugation (SC−), Single-Layer Colloidal Centrifugation (CC−), and Filtration (with SpermFilter^®^) (F−), all in extenders without antibiotics. Sperm motility, viability, and microbial load on three culture media were assessed. No significant differences were observed in the main in the sperm quality parameters among the four protocols post-processing and at 48 h (*p* < 0.05 or *p* < 0.1). Microbial loads in Columbia 5% Sheep Blood Agar and Schaedler vitamin K1 5% Sheep Blood Agar mediums were significantly higher (*p* < 0.10) for raw semen than for CS+, CC−, and F− post-processing. For Sabouraud Dextrose Agar medium, the microbial load was significantly higher (*p* < 0.10) in raw semen compared to CS+ and F−. No significant differences (*p* < 0.10) were found in 48 h chilled samples. Regardless of antibiotic presence, the evaluated processing methods, when combined with rigorous hygiene measures, maintained semen quality and reduced microbial load to the same extent as a traditional protocol using antibiotics.

## 1. Introduction

The stallion’s external genital tract exhibits a diverse microbial community, comprising both saprophytic and potentially pathogenic microbiota [1,2]. The different bacteria are constituents of the normal flora on the external genitalia of the stallion, such as *Streptococcus equisimilis*, *Streptococcus zooepidemicus*, *Bacillus* ssp., *Staphylococcus aureus*, *Escherichia coli*, *Streptococcus equi* ssp. *Zooepidemicus*, *Pseudomonas* ssp., and *Klebsiella* ssp. [3]. Although these organisms rarely produce clinical disease in the stallion, they can be transmitted to the mare’s genital tract at the time of breeding or within the insemination dose, potentially resulting in infectious endometritis and associated subfertility [4]. Moreover, excessive microbial growth has been correlated with decreased semen quality and preservation during storage [5], diminished fertility rates, and increased susceptibility to reproductive issues in mares [3]. In addition, it is widely recognized that any disturbance in the equilibrium of the normal bacterial flora of this area can create opportunities for colonization by potentially pathogenic bacteria, such us *Pseudomonas aeruginosa* and *P. pneumonia* [6].

While the addition of antibiotics in semen extenders is mandatory for trade within the European Union in some species, such a mandate does not apply for donor equines, as stipulated in the Commission Delegated Regulation (EU) 2020/686 [7]. However, it is important to note that the incorporation of antimicrobials or antimicrobial mixtures into semen diluents is a well-established and commonly adopted practice within the equine reproduction industry, and serves as a preventive tool when preparing semen doses for artificial insemination. Hence, the uterine microbiome of the mare is exposed to antibiotics during each insemination, as well as other more external parts of the reproductive tract and the environment, as a volume of the administered dose is often expelled from the uterus through retrograde flow in the subsequent hours. This process exposes the normal bacteria within the vagina and the surrounding environment to the antimicrobials agents present in the semen extenders [8]. Such exposure may contribute to increase the development of antimicrobial resistances (AMR) in these bacteria populations, which could subsequently lead to their transmission to human-associated bacteria. AMR refers to the ability of microorganisms to survive or thrive in the presence of a concentration of an antimicrobial agent which is typically effective in inhibiting or eradicating microorganisms of the same species [9]. 

The World Health Organization (WHO) has declared AMR to be one of the top ten global public health threats. The misuse and overuse of antimicrobials are the main drivers in the emergence of drug-resistant pathogens. The cost of AMR to the economy is substantial. In addition to increased mortality and morbidity, AMRs are related to extended illness, which leads to prolonged hospitalizations, an increased demand of more expensive medicines, and financial challenges for the individuals affected. Furthermore, the absence of effective antimicrobials puts the success of modern medicine in managing infections, including in vital aspects like major surgery and cancer chemotherapy, at risk. Additionally, it is important to highlight the concern from both regulatory and public health perspectives regarding the use of antimicrobials in food-producing animals, such as swine production [10] or dairy cows [11], among others. Antimicrobial resistance resulting from the agricultural use of antibiotics may impact the treatment of diseases affecting the human population that require antibiotic intervention, becoming a significant global public health concern.

In line with this approach, the European Union has established regulations regarding the use of veterinary drugs (European Regulation (EU) 2019/6). This regulation dictates that antimicrobial drugs should not be routinely employed or used to compensate for poor hygiene procedures, inadequate animal husbandry or care, or livestock farm mismanagement [9]. While semen extenders are not classified as veterinary drugs, it is imperative to follow these regulations, which emphasize the reserve use of antimicrobial medicinal products for metaphylactic purposes only when there is a high risk of infection or dissemination of an infection or infectious disease within an animal group, and suitable alternatives are not readily available.

Consequently, it is important to dedicate substantial effort to exploring different management and optimization techniques to improve semen quality while helping to reduce the microbial load. Several practices have been studied as alternatives to the incorporation of antibiotics in semen extenders [12,13,14]. In this regard, it is noteworthy to emphasize the utilization of physical techniques that can be used to improve seminal quality and microbial load. This is the case of Filtration [15,16], which has been demonstrated to reduce the number of colony-forming units (CFUs) in equine semen, Single-Layer Centrifugation [17,18,19], which has also been proven to significantly reduce bacterial load in equine semen, and the application of antimicrobial peptides [20], which have been shown to produce a significant reduction in the bacterial growth of boar semen. A diverse array of substances sourced from various origins has also been described due to their efficacy in combating microbial growth [12,13,14]. Nevertheless, to the best of our knowledge this is the first study comparing the effect of simple centrifugation, single layer colloidal centrifugation, and filtration processing on the microbial load of equine semen.

The aim of this study was to assess the impact of four different chilled semen processing protocols that include Simple Centrifugation, Single-Layer Centrifugation, and Filtration on the quality (sperm motility and viability) and microbial load of equine semen both in fresh semen (immediately after processing) and after 48 h refrigeration at 5 °C. The experiment involved comparing the use of extenders with and without antibiotics, with the purpose of potentially eliminating the prophylactic use of antibiotics in semen extenders in the future.

## 2. Materials and Methods

### 2.1. Animals

A total of ten adult stallions of different breeds (six Purebred Spanish horses, two Purebred Arabian horses, one Breton horse, and one Hispanic Arab horse) from the Ávila Horse Breeding Military Center (Centro Militar de Cría Caballar, CCFAA), which belongs to the Spanish Ministry of Defense, were used during the breeding season (March–June 2022). During the study, animals were individually housed at either Complutense Veterinary Clinical Hospital (40.44° N 3.74° W) or Ávila Horse Breeding Military Center (CCFAA) (40.66° N 4.70° W) and kept under controlled feeding and housing conditions and optimal welfare. All donor stallions, aged between 7 and 24 years, were clinically healthy and with documented fertility from artificial insemination.

### 2.2. Semen Collection and Hygiene Conditions

Semen was collected by allowing the stallions to mount a phantom and ejaculate into a Missouri-model artificial vagina (Nasco, Fort Atkinson, WI, USA) warmed to 45–50 °C and lubricated with a sterile non-spermicidal gel (Priority Care^®^, IMV Technologies, L’Aigle, France) prior to collection. A mare in estrous was used as sexual stimulation. Stallions were on a regular collection frequency basis of three collections/week during the breeding season.

A total of 10 ejaculates were used (one per stallion). Gel-free ejaculates were immediately transported to the laboratory and maintained at 37 °C for evaluation and processing.

To avoid environmental contamination, during semen collection and processing, a strict protocol that maximizes hygiene measures was implemented. All disposable materials were sterile and non-disposable materials were subjected to sterilization using an autoclave or ultraviolet light. All protocols were performed in an aseptic laboratory environment equipped with a Bunsen burner. Sterile gloves were used when preparing the artificial vagina and washing the penis with warm water followed by drying with gauzes prior to sample collection.

### 2.3. Experimental Design

Aliquots (4) of each ejaculate were processed following four different procedures (Figure 1): as the control protocol, Simple Centrifugation in extender with antibiotics (Equiplus^®^ with gentamicin, Minitüb, Tiefenbach, Germany) (SC+: 10 mL of extended semen, 1:1, *v*:*v*; 450× *g* for 7 min); Simple Centrifugation in extender without antibiotics (Equiplus^®^ without antibiotics), (SC−: 10 mL of extended semen, 1:1, *v*:*v*; 450× *g* for 7 min); Single-Layer Colloidal Centrifugation in extender without antibiotic (CC−: 10 mL of extended semen, 1:1, *v*:*v*; over 10 mL of Bottom Layer Equipure^®^, 300× *g* for 20 min); and Filtration (SpermFilter^®^) in extender without antibiotics (F−: 8 mL of extended semen, 1:1, *v*:*v*). 

After each protocol, semen samples were adjusted to 25 × 10^6^ spermatozoa/mL in Equiplus^®^ with (SC+) or without antibiotics (SC−, CC−, and F−). Each sample was then slowly chilled in a system that kept them in a water bath and maintained at 5 °C in a fridge for 48 h. Semen quality and microbial load were evaluated initially in raw semen (initial semen evaluation), immediately after the different processing procedures (post-processing evaluation in fresh sperm), and after 48 h refrigeration at 5 °C (post-chilled evaluation).

### 2.4. Semen Processing Techniques

Simple centrifugation: in 15 mL tubes, 10 mL of extended semen (1:1, *v*:*v*) in either Equiplus^®^ with gentamicin, as Control Protocol (SC+), or Equiplus^®^ without antibiotic (SC−) were centrifuged (EBA 21 Centrifuge, Hettich^®^, Hong Kong, China) at 450× *g* for 7 min [17]. 

Single-Layer Colloidal Centrifugation: in 50 mL Falcon tubes, 10 mL of extended semen (1:1, *v*:*v* in Equiplus^®^ extender without antibiotic, CC−) was pipetted and carefully layered, to avoid phase mixing, over 10 mL of Equipure^®^ (Nidacon, International AB, Mölndal, Sweden), equilibrated at 22 °C, and centrifuged (EBA 21 Centrifuge, Hettich^®^) at 300× *g* 20 min [17]. 

After both simple and colloidal centrifugation, the supernatant was removed by aspiration using a sterile disposable Pasteur pipette, leaving a percentage of seminal plasma of approximately 5–10%, and the resulting sperm pellets were resuspended to reach a final concentration of 25 × 10^6^ spermatozoa/mL. 

Filtration: 8 mL of extended semen (1:1, *v*:*v* in Equiplus^®^ extender without antibiotic, F−) was placed on the filter (SpermFilter^®^, Botucatu, Brazil). Slight movements with approximately 30 degrees of inclination were made by touching a 15 cm Petri dish as described by Alvarenga et al. [15]. After filtration, the membrane-retained spermatozoa were recovered by inverse washing of the membrane in the required volume of corresponding Equiplus^®^ without antibiotics to reach a final concentration of 25 × 10^6^ spermatozoa/mL. 

### 2.5. Cooling Processing

Immediately after processing as mentioned in Section 2.4, all samples were slowly chilled in a system that kept them in a water bath and maintained at 5 °C for 48 h. 

### 2.6. Semen Quality Evaluation

Gel-free ejaculate volume and sperm concentration were, respectively, evaluated in raw semen by using a graduated test tube and a photometer (SMD1 interspecies, Minitube^®^, Minitüb GmbH, Tiefenbach, Germany).

Motion characteristics were evaluated using a Computer-Assisted Sperm-motion Analyzer (CASA) microscope (Sperm Class Analyzer^®^, Microptic SL, Barcelona, Spain) equipped with a heated stage and phase contrast optics (X 20 objective, Optiphot-2, Nikon, Japan). Motion analysis was performed on a 10 μL drop of extended semen (25 × 10^6^ sperm/mL) placed on a preheated glass slide with a 22 × 22 mm cover slip, in at least eight fields (and at least 1000 spermatozoa in total). Experimental endpoints included total motility (TMOT, %), progressive motility (PMOT, %), velocity of the average path (VAP, μm/s), curvilinear velocity (VCL, μm/s), straight line velocity (VSL, μm/s), straightness (STR, %), linearity (LIN, %), wobble (WOB, %), lateral head displacement (ALH, μm), and beat cross frequency (BCF, Hz). The major settings used for CASA were as follows [15]: STR threshold for progressive motility, 60%; LIN threshold for circular spermatozoa, 50%; 32 frames per sequence; minimum of 15 frames per object; minimum area for objects 25 pix; and 10 mm/s as velocity limit for immobile objects

Sperm viability was determined by the eosin–nigrosin staining assay [21], mixing an aliquot of semen with tempered eosin–nigrosine on a prewarmed slide. A smear was then made using the slide-to-slide technique, drying the slide over a warm plate. The smear was subsequently fixated with the Eukitt^®^ preparation (ORSAtec, GmbH, Bobingen, Germany). A total of 200 spermatozoa were counted under a 40× magnification microscope evaluating the proportion of dead (stained) or live (unstained) spermatozoa.

Both sperm motion characteristics and vitality were performed on raw semen, immediately post-processing, and on 48 h chilled semen.

### 2.7. Microbial Load Evaluation

Aliquots (600 µL) of raw semen (RAW), immediately processed semen samples (SC+, SC−, CC−, and F−), and chilled semen aliquots (48SC+, 48SC−, 48CC−, and 48F−) were sent to the Department of Microbiology of the Veterinary Faculty, Complutense University of Madrid, for microbial quantification. 

A volume of 100 μL of each sample (dilution 0) was diluted in 900 μL of PBS solution (dilution −1) and a standardized inoculum of 100 μL was plated by spread-plate onto three different culture media: Columbia 5% Sheep Blood Agar (COS; Difco™, BD Diagnostics, Schwechat, Austria), Sabouraud Dextrose Agar (SDA; Difco™, BD Diagnostics, Schwechat, Austria), and Schaedler vitamin K1 5% Sheep Blood Agar (SCH; Difco™, BD Diagnostics, Schwechat, Austria). The plates were incubated at 37 °C for 24 h under aerobic conditions for COS and SDA mediums and under anaerobic conditions with a BD GasPak EZ System (BD Diagnostic) for SCH. Growth on agar plates was examined and colony-forming units (CFU)/mL were calculated for each culture media (microbial load).

Aliquots of the extenders with and without antibiotics were also cultured as described above, as system controls.

### 2.8. Statistical Analysis

Normality in the distribution of variables was checked using the Shapiro–Wilk test. Depending on the distribution of variables, parametric (ANOVA) or non-parametric (Kruskal–Wallis) tests were applied to analyze the data. Post hoc tests (Duncan test or Bonferroni correction for multiple comparisons, according to the distribution of variables) were performed to assess the differences between treatments (SC+, SC−, CC−, F−) and between treatments and raw semen. Data were presented as mean, median, and standard deviation values. Significant differences were considered when *p* ≤ 0.05, and *p* < 0.10 when significance values were adjusted by the Bonferroni correction, K = 6. Data were processed using the SPSS-29 statistical package (IBM^®^ SPSS^®^ Statistics, Chicago, IL, USA).

## 3. Results

### 3.1. Semen Quality

Values (mean ± SD) for initial semen quality (raw semen) are shown in Table 1. Volume, concentration, TMOT, PMOT, and viability were within the normal equine sperm range. 

In fresh sperm, immediately after processing, no significant differences were found between any of the protocols and raw semen for the evaluated sperm quality characteristics (Table 2), except for the straightness index (STR), which was significantly higher in the samples subjected to filtration without antibiotics (F−, 83.66%) and colloidal centrifugation without antibiotics (CC−, 81.54) than in raw semen (71.36%) (*p* < 0.05).

When evaluating sperm quality variables of 48 h chilled semen samples, no significant differences were found between the four processing protocols (Table 3), except for the Linearity index (LIN), which was significantly (*p* < 0.05) higher in CC− (51.5%) than in SC+ (39.49%) and F− (43.13%), and the Oscillation index (WOB), which was significantly (*p* < 0.10) higher in CC− (58.32%) than in SC+ (46.80%).

### 3.2. Microbial Load

The results for microbial load in raw semen and immediately after processing are shown in Table 4. In both Columbia 5% Sheep Blood Agar (COS) and Schaedler vitamin K1 5% Sheep Blood Agar (SCH) culture mediums, total microbial load values, expressed as colony-forming units/mL (CFU/mL), were significant higher (*p* < 0.10) for raw semen than for CS+, CC−, and F−. For these media, the samples processed by simple centrifugation without antibiotics showed a microbial load not significantly different from that of the other processing protocols. Regarding SDA medium, microbial load was only significantly higher (*p* < 0.10) when comparing raw semen to CS+ and F− processed samples. 

When evaluating bacteriology in 48 h chilled semen samples, no significant differences were found between the four processing protocols for any of the culture media evaluated (Table 5).

## 4. Discussion

Our aim was to assess the impact of four different chilled semen processing protocols that include Simple Centrifugation, Single-Layer Centrifugation, and Filtration on the quality (sperm motility and viability) and microbial load of equine semen both in fresh semen (immediately after processing) and after 48 h refrigeration at 5 °C. 

In our study, raw semen quality parameters fall well within the established norms for the equine population [22]. The mean values, represented as mean ± standard deviation, for ejaculates of all ten stallions were as follows: 83.41 ± 15.85% for total motility, 62.75 ± 22.22% for progressive motility, 82.5 ± 13.99% for percentage of live sperm, and 209.80 × 10^6^ ± 91.52 spermatozoa/mL for concentration. 

We observed a reduction in sperm quality after refrigeration, which is considered normal, as it is well-established that decreasing temperature can induce changes that may have a detrimental impact on semen quality and fertility [23,24,25]. 

Immediately after processing, the only statistically significant difference was observed in the samples subjected to filtration without antibiotics (F−) and colloidal centrifugation without antibiotics (CC−), which showed a higher STR than raw semen. In the case of 48 h chilled samples, no significant differences of particular importance were found between protocols, except for LIN and WOB. When studying the impact of the different protocols on semen quality throughout the entire refrigeration process, it can be concluded there were no important significant differences between the traditional processing protocol (CS+) and the other three evaluated procedures on the analyzed variables, neither immediately after processing nor after 48 h of refrigeration at 5 °C. Hence, semen quality in both immediately processed and 48 h chilled samples was similar irrespective of the processing method and the presence or absence of antibiotics. This observation suggests that semen quality can be effectively maintained even in the absence of antibiotics.

It may be somewhat unexpected that we did not observe a discernible improvement in semen quality when using colloidal centrifugation or filtration techniques. Historically, these processing methods have exhibited a capacity to significantly enhance equine semen quality, particularly when compared to conventional simple centrifugation [15,16,17,18,26]. Nevertheless, some authors, in agreement with our results, have not observed differences between samples processed via simple centrifugation and those subjected to colloidal centrifugation, either immediately after processing [5] or following cryopreservation [27]. Al Kass et al. [19,28], while identifying improved post-chilling quality in samples subjected to single-layer centrifugation compared to those left untreated, also reported no significant disparities between treatments with or without antibiotics. Similarly, in the case of filtration, our findings are consistent with those reported by previous studies. Alvarenga et al. [16] did not observe statistically significant differences in motility parameters when comparing filtered samples, only extended or simple centrifuged samples, both immediately after processing and after 24 h in refrigeration. Similar results were reported by other authors [15,29], who found no significant disparities in sperm quality between filtered and simple-centrifuged samples after processing or refrigeration. These distinctions were also found to be absent between these processing methods in cryopreserved samples [15]. Neto et al. [29] identified differences in post-cooling sperm quality between filtered and simple centrifuged samples when separately assessing good and bad coolers.

Nevertheless, Roach et al. [30] reported superior results for total and progressive motility (%) in samples subjected to colloidal centrifugation compared to filtration. However, our study yielded similar results for both CC and F protocols in comparison to SC, both immediately post-processing and after 48 h of refrigeration. Various potential explanations for this observation have been considered.

Firstly, most of the previously mentioned studies [5,16,17,18,26,29,30] employed extenders containing antibiotics in all protocols, which differs from our antibiotic-free procedures. It is plausible that the exclusion of antimicrobials in the colloidal centrifugation and filtration procedures in our experiment may have attenuated the expected enhancement in semen quality.

Secondly, when evaluating each horse individually we found a distinct response between treatments, which could reflect the characteristic variability in individual semen responses observed within the equine population [22,24,31]. This individual variability among stallions is mainly a result of the type of selection to which they have been subjected, based on pedigree, conformation, and performance record rather than reproductive efficiency [32]. In our study we distinguished between what we may classify as “poor” and “good” coolers. In fact, the progressive motility of conventionally processed samples (CS+) and those refrigerated for 48 h ranged from 1.7% to 61.1%. Traditionally, stallions can be classified depending on their sensitivity to the seminal refrigeration process, in terms of the effect of this process on their sperm motility, into “Bad coolers”, when they show less than 30% in total motility after cooling, and “Good coolers” if they show more than 30% in total motility [29]. This individual variability and individual semen response may account for the absence of significant differences between the protocols. When examining the effects on an individual basis, we have confirmed an enhancement in semen quality, particularly in terms of increased progressive motility, in numerous of the samples subjected to colloidal centrifugation (CC−) as compared to those processed via standard simple centrifugation (CS+). Specifically, six out of ten of the stallions exhibited at least 5 points higher progressive motility values immediately after processing with colloidal centrifugation versus the conventional protocol (CS+) (data included in Appendix A). This proportion decreased in one stallion (five out of ten) when evaluating the 48 h chilled samples (data included in Appendix B).

It is important to mention the enormous variability of factors influencing sperm quality. This is the case for the fatty acid composition of the sperm membrane, which has been proven to influence sperm motility parameters and fertility in stallion semen [33]. In this regard, these authors analyzed the liposomes of the sperm membrane and the relation to the dietary regimen, highlighting the importance of polyunsaturated acid balance and demonstrating a positive correlation between oleic acid and progressive motility in equine semen.

Regarding microbial load, in all instances, raw semen samples exhibited microbial growth in at least one of the culture media used. It is well established that following collection and processing, equine semen is non-sterile and carries a significant prevalence of contaminating bacteria originating from the external genitalia and the surrounding environment [1,2,4,34,35,36,37]. The quantity and type of this microbial flora are highly variable [34,35], and are influenced by numerous factors. Its presence was not dismissed despite the rigorous hygiene procedures applied during collection and processing, as has been previously stated by others [2,35,38].

Our results are in agreement with others who previously reported that the semen micro-organism population is diverse, encompassing both aerobic and anaerobic bacteria and fungi [2,27,35,39,40]. While we did not conduct microorganism identification in this study, we employed three specific culture media with the aim of comprehensively capturing the spectrum of normal equine semen microorganisms. Columbia 5% Sheep Blood Agar was used for the isolation of both non-fastidious and fastidious microorganisms. This primary insolation medium supports the growth of a wide range of microorganisms, such as Enterobacteriaceae, Pseudomonas, other non-fermenting Gram-negatives bacilli, *Streptococci*, *Enterococci*, *Staphylococci*, *Coryneforms*, *Candida* spp., and numerous others. Sabouraud-4% Dextrose Agar (SDA) is usually utilized as a complex medium for the culture and isolation of yeasts and mold. It is also employed as an absence test for Candida albicans. The high dextrose concentration, in addition to the low pH of this medium, fosters the growth, spore formation (*Conidia* and *Sporangia* spp.), and pigment production in yeasts and molds while inhibiting bacterial growth. Schaedler vitamin K1 5% Sheep Blood Agar (SCH) was chosen for the isolation of fastidious anaerobic bacteria and is a standard method for the isolation of strict anaerobes, including *Bacteroides*, *Prevotella*, *Porphyromonas*, *Fusobacterium*, *Clostridium*, *Peptostreptococcus*, non-forming bacilli, strictly anaerobic spores (e.g., genus *Eubacterium*), *Mobiluncus*, *Actinomyces*, and many others. For future investigations, it would be of interest to perform a microorganism isolation through conventional culture or MALDI-TOF MS to explore individual variations in bacterial taxa in equine semen or to identify commensal and potentially pathogenic bacteria using the Ion 16S™ Metagenomics Workflow [34]. 

In our study, with the exception of SC−, all protocols (SC+, CC−, and F−) significantly reduced microbial load compared to raw semen immediately after processing in fresh sperm samples, in both in COS and SCH culture media. While standard centrifugation without antibiotics did not significantly reduce the load compared to raw semen, it did not significantly differ from the other protocols. In the case of SDA media, only the control protocol (SC+) and filtration (F−) significantly reduced this load, and this leads to the belief that filtration could be more effective in reducing the yeast and fungi load. When comparing the four evaluated protocols in fresh sperm, no significant differences were found in any of the culture media. In 48 h chilled samples, there were no significant differences in microbial load in any of the culture media between protocols. Regardless of the presence of antibiotics or the type of processing, we did not find differences in bacterial growth in 48 h chilled samples. This finding is highly encouraging when considering the potential implementation of semen processing methods such as colloidal centrifugation or filtration as alternatives to the use of antibiotics in extenders. 

Varela et al., 2018 [5] also observed a significant decrease in bacterial growth in samples subjected to both standard and single-layer colloidal centrifugation compared to extended samples alone, both immediately after processing and after 48 h of refrigeration. Similar to our findings, they did not find differences between standard and single-layer centrifuged samples at either time point. It is worth noting that Varela and colleagues used an extender with antibiotics in their study. On the other hand, several authors have reported a positive effect of colloidal centrifugation in reducing the bacterial load in equine semen [12,19,27,39] and demonstrated a reduction in microbial load through single-layer centrifugation in samples both with and without antibiotics. Guimaraes et al. [27] found a lower bacterial load in frozen–thawed samples subjected to colloidal centrifugation compared to those processed by standard centrifugation. Alvarenga et al. [15] also reported a significant reduction in bacterial growth in semen using filtration.

We have hypothesized that the disparities in our results could be attributed to the rigorous hygiene procedures employed during semen collection and processing, which likely resulted in a minimal microbial load. In this context, Morrell et al. [41] noted that the efficacy of colloidal centrifugation in reducing microbial load is diminished when applied to semen samples with a low initial load, which may be the case in our case. Rather than relying solely on antibiotics, our investigation aligns with the perspective described by Schulze et al. [13], emphasizing the paramount importance of robust and comprehensive hygiene management in artificial insemination centers as a pivotal approach for controlling bacterial growth, reducing the dependance on antibiotics, and mitigating the development of antimicrobial resistances. The implementation of rigorous personnel laboratory standards and hygiene practices during semen processing have been demonstrated to curtail the introduction of environmentally associated bacteria [42].

While the microbial load was indeed reduced, it is imperative to note that, as previously emphasized by several authors, none of these protocols were able to completely remove all semen microorganisms [5,6,15,19,27,40]. Furthermore, in line with the observations made by different authors [39,40], the persistence of microbial load, even in the presence of antibiotics in the extenders, might be attributed to the existence of antibiotic-resistant microorganisms or a lack of effectiveness of the specific antimicrobial.

As already mentioned, the use of antibiotics in semen extenders is a well-established practice in stallion reproductive management. However, considering the growing concern over increasing antimicrobial resistance and its impact on society, we contend that the prophylactic and routine use of antibiotics should be reconsidered, reserving their application for strictly necessary cases. This perspective aligns with the evolving European legislation landscape. The Commission Delegated Regulation (EU) 2020/686 specifies requirements concerning antibiotics or combinations thereof with bactericidal activity in semen or included in semen extenders. However, in species other than bovine, ovine, caprine, and porcine species, where a list of antibiotics in diluted semen is provided, the obligation is not designated. In a similar vein, European Regulation (EU) 2019/6 on veterinary drugs dictates that antimicrobial drugs should not be routinely employed or used as a substitute for poor hygiene, inadequate animal husbandry or care, or the mismanagement of livestock farms. These two pieces of European legislation appear to present conflicting stances and it is imperative to work towards minimizing the prophylactic use of antibiotics to effectively combat antimicrobial resistance. As a result, it is crucial to allocate significant endeavors to explore different management and optimization techniques, as is the case in our investigation, aimed at finding alternatives to the use of antibiotics in semen extenders. Although our results have significant practical relevance, it is important to highlight certain limitations that may be encountered in our approach, firstly, the higher cost associated with these advanced processing techniques, as well as the increased training, processing, and handling times due to the implementation of hygiene measures. Another limitation could be the limited number of ejaculates and animals included in this study. It would be advisable to expand the sample size for future studies and increase the number of stallion groups to assess the effect on the variability of the equine population regarding seminal preservation processes.

## 5. Conclusions

Hence, in both equine fresh semen (immediately after processing) and in 48 h chilled semen samples, and regardless of the presence of antibiotics in semen extenders, these processing methods, when coupled with comprehensive management measures to prevent environmental contamination, effectively preserved semen quality and microbial load comparable to the traditional semen processing protocol that includes antibiotics. Although further studies are needed, these findings may support the adoption of diverse semen processing protocols and the optimization of hygiene measures as potential strategies to reduce or eliminate the non-therapeutic use of antibiotics in semen extenders and to regulate the development of bacterial resistance within semen collection centers.

## Figures and Tables

**Figure 1 animals-14-00179-f001:**
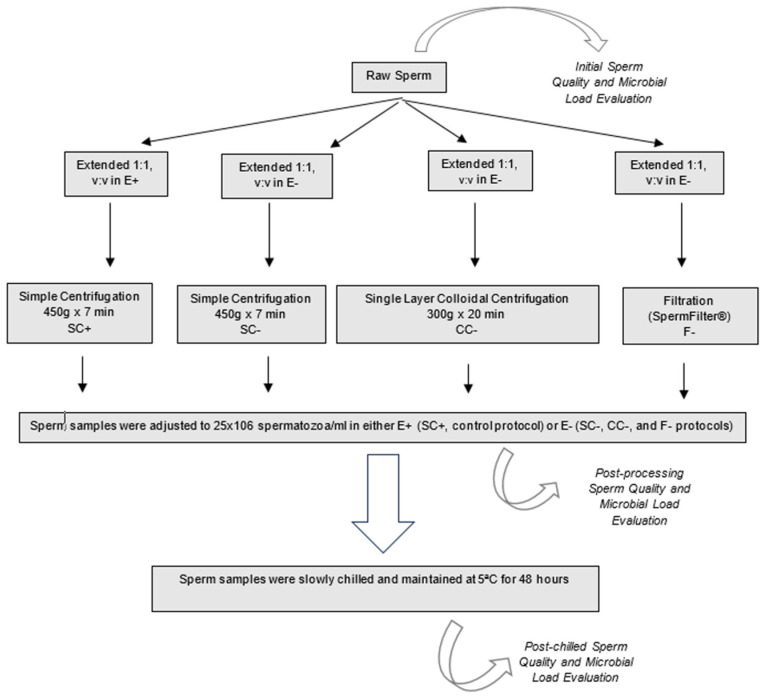
Study experimental design. E+: Equiplus^®^ with gentamicin, E−: Equiplus^®^ without antibiotics. SC: simple centrifugation (450× *g* for 7 min) of 10 mL of extended semen with (+) and without (−) antibiotics. CC−: colloidal centrifugation (300× *g* for 20 min) of 10 mL of extended semen without antibiotics over 10 mL of Bottom Layer Equipure^®^ and F−: filtration of 8 mL of extended semen without antibiotics in the Spermfilter^®^ (Botupharma, Botucatu, Sao Paulo, Brazil).

**Table 1 animals-14-00179-t001:** Mean values of semen characteristics of raw semen.

Variable	Mean ± Std. Deviation	Equine Normal Range
Volume (mL)	59.50 ± 32.77	30–100 (without gel fraction)
Concentration (×10^6^ sperm/mL)	209.80 ± 91.52	100–350
TMOT (%)	83.41 ± 15.85	
PMOT (%)	62.75 ± 22.22	Min 50%
Lives (%)	82.50 ± 13.99	>65%

Means values (mean ± SD) of initial volume (mL), concentration (×10^6^ spermatozoa/mL), percentage of total motile sperm (%; MOT), percentage of progressively motile sperm (%; PMOT), and percentage of lives spermatozoa (%, Lives) in raw semen.

**Table 2 animals-14-00179-t002:** Mean values of sperm quality in raw semen and after initial processing (fresh sperm).

Protocol	TMOT (%)	PMOT (%)	VCL (µm/s)	VAP (µm/s)	VSL (µm/s)	STR (%)	LIN (%)	WOB (%)	ALH (µm)	BCF(Hz)	Lives (%)
RAW	83.41 ^a^ ± 15.85	62.75 ^a^ ± 22.22	101.00 ^a^ ± 27.63	76.05 ^a^ ± 19.02	55.51 ^a^ ± 14.23	71.36 ^b^ ± 10.66	55.33 ^a^ ± 13.26	74.81 ^a^ ± 9.86	3.16 ^a^ ± 1.20	8.36 ^a^ ± 1.81	82.50 ^a^ ± 13.99
SC+	80.46 ^a^ ± 17.67	61.32 ^a^ ± 20.84	94.95 ^a^ ± 16.92	72.91 ^a^ ± 16.35	57.41 ^a^ ± 14.16	75.94 ^a,b^ ± 7.77	59.20 ^a^ ± 11.93	75.37 ^a^ ± 9.75	2.85 ^a^ ± 0.56	9.27 ^a^ ± 1.18	84.60 ^a^ ± 9.21
SC−	80.23 ^a^ ± 21.57	59.13 ^a^ ± 21.84	86.41 ^a^ ± 22.97	67.39 ^a^ ± 19.48	53.66 ^a^ ± 16.01	76.70 ^a,b^ ± 9.55	60.42 ^a^ ± 12.60	75.91 ^a^ ± 8.87	2.68 ^a^ ± 0.78	8.34 ^a^ ± 1.40	85.05 ^a^ ± 9.14
CC−	85.50 ^a^ ± 12.87	63.19 ^a^ ± 18.87	79.42 ^a^ ± 17.96	63.50 ^a^ ± 15.14	53.65 ^a^ ± 12.35	** 81.54 ^a^ ± 5.46 **	66.50 ^a^ ± 8.21	79.36 ^a^ ± 7.54	2.40 ^a^ ± 0.61	8.23 ^a^ ± 1.13	84.90 ^a^ ± 13.50
F−	70.95 ^a^ ± 21.75	52.54 ^a^ ± 21.08	76.15 ^a^ ± 20.78	57.86 ^a^ ± 17.88	50.18 ^a^ ± 13.08	** 83.66 ^a^ ± 5.78 **	64.79 ^a^ ± 10.38	74.79 ^a^ ± 9.97	2.40 ^a^ ± 0.64	9.30 ^a^ ± 1.63	77.35 ^a^ ± 13.27

Mean values (expressed as mean ± Standard Deviation) of sperm total (TMOT, %) and progressive motility (PMOT, %), curvilinear velocity (VCL, μm/s), velocity of the average path (VAP, μm/s), straight line velocity (VSL, μm/s), straightness (STR, %), linearity (LIN, %), wobble (WOB, %), lateral head displacement (ALH, my), beat cross frequency (BCF, Hz), and viability (Lives, %) in raw semen (RAW) and immediately after the four processing protocols (Simple Centrifugation in extender with antibiotics, SC+; Simple Centrifugation in extender without antibiotics, SC−; Single-Layer Colloidal Centrifugation in extender without antibiotics, CC−; and Filtration in extender without antibiotics, F−. Superscript letters represent significant differences (in bold and in red) between treatments (*p* < 0.05) in each variable.

**Table 3 animals-14-00179-t003:** Mean values of sperm quality in chilled semen after 48 h at 5 °C.

Protocol	TMOT (%)	PMOT (%)	VCL (µm/s)	VAP (µm/s)	VSL (µm/s)	STR (%)	LIN (%)	WOB (%)	ALH (µm)	BCF(Hz)	Lives (%)
48SC+	36.23 ^a^ ± 23.61	18.78 ^a^ ± 21.81	61.31 ^a^ ± 20.46	28.96 ^a^ ± 11.36	24.84 ^a^ ± 11.35	84.01 ^a^ ± 6.33	39.49 ^b^ ± 6.01	46.80 ^b^ ± 4.25	2.74 ^a^ ± 0.68	10.22 ^a^ ± 3.31	60.75 ^a^ ± 16.11
48SC−	43.87 ^a^ ± 24.48	25.12 ^a^ ± 20.94	60.35 ^a^ ± 15.94	31.70 ^a^ ± 9.77	27.71 ^a^ ± 9.82	86.41 ^a^ ± 6.64	45.68 ^a,b^ ± 7.27	47.00 ^a,b^ ± 16.51	3.54 ^a^ ± 3.16	10.11 ^a^ ± 3.15	60.10 ^a^ ± 12.95
48CC−	39.77 ^a^ ± 30.07	20.71 ^a^ ± 21.32	46.21 ^a^ ± 18.63	27.40 ^a^ ± 12.55	24.44 ^a^ ± 12.48	88.03 ^a^ ± 8.17	** 51.5 ^a^ ± 8.78 **	** 58.32 ^a^ ± 6.4 **	1.80 ^a^ ± 1.03	7.63 ^a^ ± 5.04	53.15 ^a^ ± 25.26
48F−	33.12 ^a^ ± 18.61	16.24 ^a^ ± 12.56	58.86 ^a^ ± 17.51	29.60 ^a^ ± 10.85	25.55 ^a^ ± 11.19	83.81 ^a^ ± 9.72	43.13 ^b^ ± 9.17	51.12 ^a,b^ ± 6.51	2.40 ^a^ ± 0.85	9.28 ^a^ ± 4.42	57.50 ^a^ ± 23.33

Mean values (expressed as mean ± Standard Deviation) of sperm total (TMOT, %) and progressive motility (PMOT, %), curvilinear velocity (VCL, μm/s), velocity of the average path (VAP, μm/s), straight line velocity (VSL, μm/s), straightness (STR, %), linearity (LIN, %), wobble (WOB, %), lateral head displacement (ALH, μm), beat cross frequency (BCF, Hz) and viability (Lives, %) in chilled semen after 48 h at 5 °C in the four processing protocols (Simple Centrifugation in extender with antibiotics, SC+; Simple Centrifugation in extender without antibiotics, SC−; Single-Layer Colloidal Centrifugation in extender without antibiotics, CC−; and Filtration in extender without antibiotics, F−. Superscript letters represent significant differences (in bold and in red) between treatments (*p* < 0.05 for LIN and *p* < 0.10 for WOB) in each variable.

**Table 4 animals-14-00179-t004:** Total microbial load in three different culture media in raw semen and immediately after processing samples (fresh sperm).

	COS (CFU/mL)		SDA (CFU/mL)		SCH (CFU/mL)	
	Mean	Median	Mean	Median	Mean	Median
RAW	4926.00 ^a^ ± 7322.59	1085.00	86.00 ^a^ ± 185.12	15.00	511.00 ^a^ ± 1007.62	140.00
SC+	**216.50 ^b^ ± 504.41**	5.00	**1.00 ^b^ ± 3.16**	0.00	**25.00 ^b^ ± 57.59**	0.00
SC−	348.30 ^a,b^ ± 792.99	35.00	11.00 ^a,b^ ± 31.43	0.00	41.00 ^a,b^ ± 60.45	20.00
CC−	**731.30 ^b^ ± 2180.11**	10.00	4.00 ^a,b^ ± 6.99	0.00	**10.00 ^b^ ± 18.26**	5.00
F−	**51.70 ^b^ ± 64.64**	10.00	**1.00 ^b^ ± 3.16**	0.00	**26.00 ^b^ ± 42.48**	15.00

Means (mean ± SD) and median values of total microbial load, expressed as colony forming units/mL (CFU/mL)) in raw semen (RAW), Simple Centrifugation in extender with antibiotics (SC+), Simple Centrifugation in extender without antibiotics (SC−), Single-Layer Colloidal Centrifugation in extender without antibiotics (CC−), and Filtration in extender without antibiotics (F−) samples in the different culture media: Columbia 5% Sheep Blood Agar (COS), Sabouraud Dextrose Agar (SDA), and Schaedler vitamin K1 5% Sheep Blood Agar (SCH). Superscript letters represent significant differences (in bold) between treatments (*p* < 0.10) in each variable.

**Table 5 animals-14-00179-t005:** Total microbial load in three different culture media in 48 h chilled semen samples.

	COS (CFU/mL)		SDA (CFU/mL)		SCH (CFU/mL)	
	Mean	Median	Mean	Median	Mean	Median
48SC+	121.80 ^a^ ± 233.36	0.00	0.00 ^a^ ± 0.00	0.00	2.00 ^a^ ± 4.22	0.00
48SC−	250.00 ^a^ ± 658.16	10.00	8.00 ^a^ ± 22.01	0.00	27.50 ^a^ ± 52.88	0.00
48CC−	347.44 ^a^ ± 873.74	20.00	2.22 ^a^ ± 6.67	0.00	14.70 ^a^ ± 20.89	10.00
48F−	1447.70 ^a^ ± 4480.8	30.00	4.00 ^a^ ± 9.66	0.00	6.90 ^a^ ± 15.01	0.00

Means (mean ± SD) and median values of total microbial load, expressed as colony forming units/mL (CFU/mL)) in Simple Centrifugation in extender with antibiotics (48SC+), Simple Centrifugation in extender without antibiotics (48SC−), Single-Layer Colloidal Centrifugation in extender without antibiotics (48CC−), and Filtration in extender without antibiotics (48F−) samples after 48 h at 5 °C and in the different culture media: Columbia 5% Sheep Blood Agar (COS), Sabouraud Dextrose Agar (SDA), and Schaedler vitamin K1 5% Sheep Blood Agar (SCH). Superscript letters represent significant differences between treatments (*p* < 0.10) in each variable.

## Data Availability

The data presented in this study are available on request from the corresponding author. The data are not publicly available due to privacy restrictions.

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
