# Peer review of "Strategies to Reduce the Use of Antibiotics in Fresh and Chilled Equine Semen"

_animals, 2024, doi:10.3390/ani14020179_

Round 1

Reviewer 1 Report

Comments and Suggestions for Authors

Specific comments, critiques and suggestions for improvement are below.

Line 43:                 Use ‘as follows’ not ‘a follow’

Lines 55 and 56:   Use ‘significantly’ not ‘significant’

Line 130:               Use ‘eliminating’ not ‘eliminate’

Lines 140 to 141:  Was the documented fertility of the stallions from natural service, artificial insemination or both?

Lines 142 to 146:  What was the collection frequency of the stallions prior to the collection of the ejaculate used for the study? For example, were the stallions on a regular collection frequency of once per week, or once per month? Or were the stallions only collected when the ejaculate for the study was used?

Line 144:               A space needs to be put between Fort Atkinson

Lines 166 to 167:  Why were the semen samples not adjusted to 25 x 106 spermatozoa/ml before the treatments were performed? In this reviewer’s opinion, adjusting to 25 x 106 spermatozoa/ml after the treatments were performed could introduce unnecessary variability to the results. In addition, having a different spermatozoa/ml concentration due to a 1:1, v:v ratio before the treatments were applied within each treatment could introduce unnecessary variability to the results.

Line 218:               Use ‘eosin’ not ‘eosing’

Lines 242 to 251:  In the statistical analysis, it needs to be stated if repeated measures analysis were used since the samples were evaluated over time.

In addition, what was the statistical model used? For example, how was the effect of stallion accounted for in the statistical analysis (perhaps this reviewer missed this in the statistical analysis section)? Explaining how the effect of stallion was accounted for in the statistical model is important especially due to the high variability of results for individual stallions, as the authors acknowledge in the Discussion section.

Line 260:               This reviewer assumes that the term ‘Alives’ refers to live spermatozoa. In the materials and methods, the section describing how sperm viability was determined using the eosin-nigrosin staining assay, the term ‘live’ spermatozoa is used. This reviewer would recommend using ‘live’ rather than ‘alives’ to maintain consistency in the document.

There seems to be a formatting issue in the review PDF document sent to the reviewers. About halfway through the document, right after table 2, the line numbers are no longer present and the page numbering starts over as page 2 of 15. This is why a ‘?’ is used below instead of a line number.

Lines ?:                 In the second paragraph discussion section, this reviewer is confused by the sentences, “Nevertheless, in our study we distinguished between what we may classified as “poor” and “good” coolers. In fact, the progressive motility of conventionally processed samples (CS+) and refrigerated for 48 hours ranged from 1,7% to 61,1%.” Please expand on this discussion of ‘poor’ and ‘good’ coolers’ to clarify why this information is included in the discussion (or possibly reference this information in the appendix). It seems this information is discussed more in depth later in the document, particularly in the paragraph for the second explanation.

Line ?:                   Is “(data included in appendix 2)” referencing appendix B?

Comments on the Quality of English Language

None

Author Response

Line 43:                 Use ‘as follows’ not ‘a follow’

This was corrected (line 62). The comment has been taken, although now all that text has been crossed out to condense the abstract and adapt it to contain only 200 words.

Lines 55 and 56:   Use ‘significantly’ not ‘significant’

This was corrected (lines 77 and 79). The comment has been taken into account, although now all that text has been crossed out to condense the abstract and adapt it to contain only 200 words.

Line 130:               Use ‘eliminating’ not ‘eliminate’

This was corrected (line 166).

Lines 140 to 141:  Was the documented fertility of the stallions from natural service, artificial insemination or both?

From artificial insemination (this information was included in line 178).

Lines 142 to 146:  What was the collection frequency of the stallions prior to the collection of the ejaculate used for the study? For example, were the stallions on a regular collection frequency of once per week, or once per month? Or were the stallions only collected when the ejaculate for the study was used?

Stallions were on a regular collection frequency basis of three collections/week during the breeding season. This information was included in lines 183-185.

Line 144:               A space needs to be put between Fort Atkinson

This was corrected (line 181).

Lines 166 to 167:  Why were the semen samples not adjusted to 25 x 106 spermatozoa/ml before the treatments were performed? In this reviewer’s opinion, adjusting to 25 x 106 spermatozoa/ml after the treatments were performed could introduce unnecessary variability to the results. In addition, having a different spermatozoa/ml concentration due to a 1:1, v:v ratio before the treatments were applied within each treatment could introduce unnecessary variability to the results.

Samples were not adjusted to that concentration before performing the treatments because we have followed the conventional processing protocol for preparing cooled equine semen doses that require centrifugation, which involves the dilution of raw semen 1:1, v:v with the cooling extender before centrifugation and subsequent resuspension at the final dose for the descending temperature and refrigeration maintenance.

According to the recommendations for handling refrigerated equine semen, the final concentration of the dose, to maximize sperm viability should be between 25-50 x 106 spermatozoa/ml (A. Bradecamp, Chapter 126 Preparation of Semen for cooled transport, pp. 409-411. In: Equine Reproductive procedures, First Edition, Edited by Dascanio and McCue . 2014 John Wiley & Sons).

Moreover, this concentration is within the range of what is it required to perform an adequate computer-assisted sperm analysis (J.J Dascanio, Chapter 113 Computer-Assisted Sperm Analysis, pp- 368-372. In:  Equine Reproductive procedures, First Edition, Edited by Dascanio and McCue . 2014 John Wiley & Sons.)

Line 218:               Use ‘eosin’ not ‘eosing’

This was corrected (line 258).

Lines 242 to 251:  In the statistical analysis, it needs to be stated if repeated measures analysis were used since the samples were evaluated over time.

We did not perform a repeated meaures analysis over times of the variables as protocols were compared to raw sperm immediately after processing and only between them after 48h of refrigeration.

In addition, what was the statistical model used? For example, how was the effect of stallion accounted for in the statistical analysis (perhaps this reviewer missed this in the statistical analysis section)? Explaining how the effect of stallion was accounted for in the statistical model is important especially due to the high variability of results for individual stallions, as the authors acknowledge in the Discussion section.

Depending on the distribution of variables, parametric (ANOVA) or non-parametric (Kruskal Wallis) tests were applied to analyze the data. Post hoc tests (Duncan test or Bonferroni correction for multiple comparisons, according to the distribution of variables) were performed to assess the differences between treatments (SC+, SC-, CC-, F-) and between treatments and raw semen. Data were presented as mean, median and standard deviation values. Significant differences were considered when p ≤ 0.05, and p<0.10 when significance values were adjusted by the Bonferroni correction, K=6. As already mentioned, variability among stallions is a common finding that equine andrologists are used work with. In the case of the study this individual variability is accounted in the statistical analysis on the distribution of variables and standard deviation of values in each variable. We do not work with ejaculate pools, which may reduce the effect of individual variability.

Line 260:               This reviewer assumes that the term ‘Alives’ refers to live spermatozoa. In the materials and methods, the section describing how sperm viability was determined using the eosin-nigrosin staining assay, the term ‘live’ spermatozoa is used. This reviewer would recommend using ‘live’ rather than ‘alives’ to maintain consistency in the document.

This term was changed in the MS (Tables 1-3, and lines 300, 315 and 321).

There seems to be a formatting issue in the review PDF document sent to the reviewers. About halfway through the document, right after table 2, the line numbers are no longer present and the page numbering starts over as page 2 of 15. This is why a ‘?’ is used below instead of a line number.

This was corrected. We are sorry for the inconveniences.

Lines ?:                 In the second paragraph discussion section, this reviewer is confused by the sentences, “Nevertheless, in our study we distinguished between what we may classified as “poor” and “good” coolers. In fact, the progressive motility of conventionally processed samples (CS+) and refrigerated for 48 hours ranged from 1,7% to 61,1%.” Please expand on this discussion of ‘poor’ and ‘good’ coolers’ to clarify why this information is included in the discussion (or possibly reference this information in the appendix). It seems this information is discussed more in depth later in the document, particularly in the paragraph for the second explanation.                           

This information was clarified, included in the mentioned paragraph and extended also following suggestions of reviewer 2 (lines 414-423).

Line ?:                   Is “(data included in appendix 2)” referencing appendix B?

Yes, this was corrected (lines 431 and 432).

Reviewer 2 Report

Comments and Suggestions for Authors

I commend the authors for presenting such a relevant and well-performed study. As highlighted in the Introduction, antimicrobial resistance is a current issue that directly involves veterinarians. Particularly, in farm animals, although some regulations have been made to address this issue, it is still common to see the indiscriminate use of antibiotics when they are not even necessary. The present study gives proof that there are alternatives that can be used for equines that minimize/prevent the use of antibiotics without affecting semen quality. I made some suggestions hoping they are helpful for the authors.

Simple summary. This section adequately covers the general aspects of the study. As a minor recommendation to improve it, I suggest adding, as in the abstract, a very small sentence stating that antimicrobials are commonly used in the semen used for artificial insemination, possibly making it part of the worldwide issue regarding antimicrobial resistance.

Abstract. In lines 40–41, it is not clear if the ten samples were used for the four processing techniques, or if for each technique a total of ten samples was collected. Also, it might not be a big issue, but the journal’s template states that the Abstract needs a maximum of 200 words. Revise if the current Abstract is within this number (if not, try to amend by keeping the most relevant findings of the study).

Introduction. A very good background of the issue the authors are addressing. A couple of recommendations to improve this section would be, after the sentence on line 124, I would recommend adding a summary of the results that these techniques have reported to date. If the present study is the first study that compares the four techniques to assess its effect on microbial load, please, mention it.

Line 255. If the abbreviations have been previously defined (as in line 211), there is no need to define them again in the following text.

Table 1. Since it is mentioned in line 256 that values were within the normal range, I would recommend adding a column to the right stating these normal ranges for equines to improve the table.

Discussion. I’m sorry I can´t write the specific line (I couldn´t see line numbering in this section), but in the first paragraph of page 10, the authors mention that they distinguished as “poor” and “good” coolers. Then, they mention that CS+ motility ranged between 1.7–61.1%. Could the authors include if this is considered “good” or “poor”? The same observation would apply to the other techniques.
At the end of the second paragraph, it is written that semen quality was maintained even in the absence of antibiotics. Is there another study assessing the use of antibiotics in semen? If so, I recommend discussing this, even if it was made in another species, to link the present findings to previous information.
In the paragraph that starts with “Secondly, when evaluating each horse individually we found….”, I recommend clearly stating what factors can influence these different results according to the individual. For example, breed? Age? It would be interesting to discuss the age since the authors used a wide range for the stallion (7–24 years).
In the paragraph starting with “In our study, with the exception of SC-, all protocols…”, could the authors add here or in the Introduction the maximum permitted bacterial load? The authors state that the techniques reduced the load, but it would be helpful for the reader to know if there is a maximum permitted load.
I consider that the authors can shorten the last paragraph of the Discussion. From where the Commission Delegated Regulation is mentioned, this information was already included in the Introduction. I understand that the authors are trying to make the connection between current regulations and the findings; however, I consider that this information could be summarized (not deleted) without losing the importance of the current findings.  

Reference list. Please, revise the journal’s reference format and amend the list.

Author Response

I commend the authors for presenting such a relevant and well-performed study. As highlighted in the Introduction, antimicrobial resistance is a current issue that directly involves veterinarians. Particularly, in farm animals, although some regulations have been made to address this issue, it is still common to see the indiscriminate use of antibiotics when they are not even necessary. The present study gives proof that there are alternatives that can be used for equines that minimize/prevent the use of antibiotics without affecting semen quality. I made some suggestions hoping they are helpful for the authors. 

We really appreciate your comments and suggestions.

Simple summary. This section adequately covers the general aspects of the study. As a minor recommendation to improve it, I suggest adding, as in the abstract, a very small sentence stating that antimicrobials are commonly used in the semen used for artificial insemination, possibly making it part of the worldwide issue regarding antimicrobial resistance.

This suggestion was included (lines 23-24).

Abstract. In lines 40–41, it is not clear if the ten samples were used for the four processing techniques, or if for each technique a total of ten samples was collected. Also, it might not be a big issue, but the journal’s template states that the Abstract needs a maximum of 200 words. Revise if the current Abstract is within this number (if not, try to amend by keeping the most relevant findings of the study).

Following your recommendation and the opinion of another reviewer, the abstract was rewritten (lines 38-54).

Introduction. A very good background of the issue the authors are addressing. A couple of recommendations to improve this section would be, after the sentence on line 124, I would recommend adding a summary of the results that these techniques have reported to date. If the present study is the first study that compares the four techniques to assess its effect on microbial load, please, mention it.

This suggestion was included (lines 148-160).

Line 255. If the abbreviations have been previously defined (as in line 211), there is no need to define them again in the following text.

This was corrected (line 295).

Table 1. Since it is mentioned in line 256 that values were within the normal range, I would recommend adding a column to the right stating these normal ranges for equines to improve the table.

This suggesting column was included in table 1. No specific range is usually established for total motility.

Discussion. I’m sorry I can´t write the specific line (I couldn´t see line numbering in this section),

This was corrected. We are sorry for the inconveniences.

but in the first paragraph of page 10, the authors mention that they distinguished as “poor” and “good” coolers. Then, they mention that CS+ motility ranged between 1.7–61.1%. Could the authors include if this is considered “good” or “poor”? The same observation would apply to the other techniques.

Traditionally, stallions can be classified depending on their sensitivity to the seminal refrigeration process, in terms of the effect of this process on their sperm motility, into “Bad coolers”, when they show less than 30% in total motility after cooling and “Good coolers” if they show more than 30% in total motility (Neto et al., 2013). This information was included in the discussion and combined, suggested by another of the reviewers, with the information included in other paragraph (lines 414-423).

Reference: Neto, C.R.; Monteiro, G.A.; Soares, R.F.; Pedrazzi, C.; Dell’aqua, J.A.; Papa, F.O.; Alvarenga, M.A. Effect of removing seminal plasma using a Sperm Filter on the viability of refrigerated stallion semen. J. Equine Vet. Sci. 2013, 33,  40-43.

At the end of the second paragraph, it is written that semen quality was maintained even in the absence of antibiotics. Is there another study assessing the use of antibiotics in semen? If so, I recommend discussing this, even if it was made in another species, to link the present findings to previous information.

This comment has been taken into account in lines 390 to 392. Results are discussed with already included studies (references 5, 19, 28) and a new reference has been included.

New reference: Al-Kass, Z.; Spergser, J.; Aurich, C.; Kuhl, J.; Schmidt, K.; Morrell, J.M. Effect of presence or absence of antibiotics and use of modified single layer centrifugation on bacteria in pony stallion semen. Reprod. Dom. Anim. 2019,54(2),342-349.

In the paragraph that starts with “Secondly, when evaluating each horse individually we found….”, I recommend clearly stating what factors can influence these different results according to the individual. For example, breed? Age? It would be interesting to discuss the age since the authors used a wide range for the stallion (7–24 years).

Individual variability in ejaculate quality among stallions is considered the main factor that has prevented the widespread use of chilled and cryopreserved spermatozoa (Aurich, 2008; Loomis and Graham 2008). Although some variability has been reported between breeds (Alvarenga et al., 2003), it is considered mainly the result of the type of selection to which they have been subjected, based on pedigree, conformation, and performance record rather than reproductive efficiency (Varner et al., 2008). This information was included in the MS (lines 414-423).

REFERENCES:

  • Varner, D.D.; Love, C.C.; Brinsko, S.P.; Blanchard, T.L.; Bliss, S.; Carroll, S.; Eslick, M. Semen processing for the subfertile stallion. Equine Vet. Sci. 2008, 28, 677-685.
  • Alvarenga, M.A.; Leao, K.M.; Papa, F.O.; Landim-Alvarenga, F.C.; Medeiros, A.S.L.; Gomes, G.M. 2003. The use of alternative cryoprotectors for freezing stallion semen. In: Proceedings, Workshop on Transporting Gametes and Embryos, Have meyer Foundation. 2003, pp.74–76.
  • Aurich, C.; Recent advances in cooled-semen technology. Reprod. Sci. 2008,107, 268–275.
  • Loomis, P.R.; Graham, J.K. Commercial semen freezing: individual male variation in cryosurvival and the response of stallion sperm to customized freezing protocols. Reprod. Sci. 2008, 105,119–128.

In the paragraph starting with “In our study, with the exception of SC-, all protocols…”, could the authors add here or in the Introduction the maximum permitted bacterial load? The authors state that the techniques reduced the load, but it would be helpful for the reader to know if there is a maximum permitted load.

To the best of the authors' knowledge, there is no established maximum allowed bacterial load for equine semen, at least within the framework of the European Union where the authors practice the profession. Therefore, the statement that the techniques reduce the microbial load refers to their comparison to raw semen.

I consider that the authors can shorten the last paragraph of the Discussion. From where the Commission Delegated Regulation is mentioned, this information was already included in the Introduction. I understand that the authors are trying to make the connection between current regulations and the findings; however, I consider that this information could be summarized (not deleted) without losing the importance of the current findings.  

This paragraph was shortened accordingly (lines 527-534).

Reference list. Please, revise the journal’s reference format and amend the list.

This has been corrected. We apologize for any inconvenience.

Reviewer 3 Report

Comments and Suggestions for Authors

CONTEXT

The paper, titled “Strategies to reduce the use of antibiotics in equine fresh and chilled semen “ addresses an important and timely topic. I found the subject matter fascinating and read the manuscript with great interest. The paper aligns well with the scope of the journal. It addresses a specific gap in the field by suggesting the potential of alternative semen processing protocols, coupled with strict hygiene practices, to mitigate or eliminate the nontherapeutic use of antibiotics and contribute to the control of antimicrobial resistances (AMR). However, I believe that in its current form, it has several shortcomings.

BRIEF SUMMARY OF THE PAPER

The aim of this study was to assess the impact of four different chilled semen processing protocols that include Simple Centrifugation, Single-Layer Centrifugation and Filtration on the quality (sperm motility and viability) and microbial load of equine semen both in fresh semen (immediately after processing) and after 48 hours in refrigeration at 5ºC. A total of ten adult stallions of different breeds were used during the breeding season (March–June 2022). The experiment involved comparing the use of extenders with and without antibiotics, with the purpose of potentially eliminate the prophylactic use of antibiotics in semen extenders in the future.

SIMPLE SUMMARY

The simple summary is clearly written, using appropriate terms that enable understanding of the text even by a non-expert audience.

ABSTRACT

The abstract correlates with the manuscript content, but I recommend rewriting this section including more results and the significance of the obtained data.

KEYWORDS

The keywords are suitable, as they do not include terms that are already present in the article title.

INTRODUCTION

The introduction is clear and outlines the context of the study.

However, I suggest citing the following article: 10.1016/j.rvsc.2023.03.008. This is to expand the section on antibiotic use in animal husbandry, reporting additional studies dealing with this issue in other animal species.

MATERIALS AND METHODS

I suggest expanding the section 2.1. Animals, adding more detailed information about the subjects involved in the study.

The description of the procedures is detailed and comprehensive.

RESULTS

The section is complete; it clearly lays out the data obtained. However, I recommend graphically highlighting the significance of the data in Table 3 and Table 4, as it is hardly visible when reading the results.

DISCUSSION

Starting the discussion section by reiterating the aim of the study can provide clarity and context for readers.

I kindly suggest expanding the discussion section of your paper to include practical applications and a thorough exploration of the study's limitations. This addition will enhance the overall value of your research and provide a more comprehensive understanding of its implications.

I suggest that the authors consider discussing the use of lipidomics techniques as a future perspective, citing the following article: 10.3390/ANI13010008.

CONCLUSION

The conclusions are consistent with the evidence and arguments presented and they address the main question posed.

REFERENCES

All the references are included in the main text.

EDITING

There are some editing issues. It's recommended to thoroughly review the document for such problems.

Author Response

The paper, titled “Strategies to reduce the use of antibiotics in equine fresh and chilled semen “ addresses an important and timely topic. I found the subject matter fascinating and read the manuscript with great interest. The paper aligns well with the scope of the journal. It addresses a specific gap in the field by suggesting the potential of alternative semen processing protocols, coupled with strict hygiene practices, to mitigate or eliminate the nontherapeutic use of antibiotics and contribute to the control of antimicrobial resistances (AMR). However, I believe that in its current form, it has several shortcomings.

We really appreciate your comments and suggestions.

BRIEF SUMMARY OF THE PAPER

The aim of this study was to assess the impact of four different chilled semen processing protocols that include Simple Centrifugation, Single-Layer Centrifugation and Filtration on the quality (sperm motility and viability) and microbial load of equine semen both in fresh semen (immediately after processing) and after 48 hours in refrigeration at 5ºC. A total of ten adult stallions of different breeds were used during the breeding season (March–June 2022). The experiment involved comparing the use of extenders with and without antibiotics, with the purpose of potentially eliminate the prophylactic use of antibiotics in semen extenders in the future.

SIMPLE SUMMARY

The simple summary is clearly written, using appropriate terms that enable understanding of the text even by a non-expert audience.

ABSTRACT

The abstract correlates with the manuscript content, but I recommend rewriting this section including more results and the significance of the obtained data.

Following your recommendation and the opinion of another reviewer, the abstract was rewritten (lines 38-53).

KEYWORDS

The keywords are suitable, as they do not include terms that are already present in the article title.

INTRODUCTION

The introduction is clear and outlines the context of the study.

However, I suggest citing the following article: 10.1016/j.rvsc.2023.03.008. This is to expand the section on antibiotic use in animal husbandry, reporting additional studies dealing with this issue in other animal species.

We appreciate your suggestion in order to expand this section in regard to antibiotics use and its problems among animal husbandry. We have read the suggested article, “Health and welfare assessment of beef cattle during the adaptation period in a specialized commercial fattening unit”, and although of much interest, and we have found difficulties in adapting its content to our actual topic. In order to take into account your suggestion we have included two references (lines 130-136).

MATERIALS AND METHODS

I suggest expanding the section 2.1. Animals, adding more detailed information about the subjects involved in the study.

As suggested, more information was included (lines 170-178).

The description of the procedures is detailed and comprehensive.

RESULTS

The section is complete; it clearly lays out the data obtained. However, I recommend graphically highlighting the significance of the data in Table 3 and Table 4, as it is hardly visible when reading the results.

As suggested, and in order to highlight the significance of the data, significantly values were present in red (tables 3 and 4).

DISCUSSION

Starting the discussion section by reiterating the aim of the study can provide clarity and context for readers.

As suggested, we included in the start of our discussion the aim of our study. 355-358.

I kindly suggest expanding the discussion section of your paper to include practical applications and a thorough exploration of the study's limitations. This addition will enhance the overall value of your research and provide a more comprehensive understanding of its implications.

As suggested, we included this request (lines 536-544).

I suggest that the authors consider discussing the use of lipidomics techniques as a future perspective, citing the following article: 10.3390/ANI13010008.

As suggested, we included in lines 536-544.

CONCLUSION

The conclusions are consistent with the evidence and arguments presented and they address the main question posed.

REFERENCES

All the references are included in the main text.

EDITING

There are some editing issues. It's recommended to thoroughly review the document for such problems.

This was corrected. We are sorry for the inconveniences.

Round 2

Reviewer 3 Report

Comments and Suggestions for Authors

The authors have diligently addressed the review comments, significantly enhancing the paper's quality. As a result, it is now well-suited for publication.